# Rates and causes of mortality among children and young people with and without intellectual disabilities in Scotland: a record linkage cohort study of 796 190 school children

Gillian S Smith ⬤ , Michael Fleming, Deborah Kinnear, Angela Henderson ⬤ , J P Pell, Craig Melville, Sally-Ann Cooper ⬤

Institute of Health and Wellbeing, University of Glasgow, Glasgow, UK

**Correspondence to**
Professor Sally-Ann Cooper;
Sally-Ann.Cooper@glasgow.ac.uk

## ABSTRACT

**Objectives** To investigate mortality rates and causes in children and young people with intellectual disabilities.
**Design** Retrospective cohort; individual record linkage between Scotland's annual pupil census and National Records of Scotland death register.
**Setting** General community.
**Participants** Pupils receiving local authority-funded schooling in Scotland, 2008 to 2013, with an Additional Support Need due to intellectual disabilities, compared with other pupils.
**Main outcome measures** Deaths up to 2015: age of death, age-standardised mortality ratios (age-SMRs); causes of death including cause-specific age-SMRs; avoidable deaths as defined by the UK Office of National Statistics.
**Results** 18 278/947 922 (1.9%) pupils had intellectual disabilities. 106 died over 67 342 person-years (crude mortality rate=157/100 000 person-years), compared with 458 controls over 3 672 224 person-years (crude mortality rate=12/100 000 person-years). Age-SMR was 11.6 (95% CI 9.6 to 14.0); 16.6 (95% CI 12.2 to 22.6) for female pupils and 9.8 (95% CI 7.7 to 12.5) for male pupils. Most common main underlying causes were diseases of the nervous system, followed by congenital anomalies; most common all-contributing causes were diseases of the nervous system, followed by respiratory system; most common specific contributing causes were cerebral palsy, pneumonia, respiratory failure and epilepsy. For all contributing causes, SMR was 98.8 (95% CI 69.9 to 139.7) for congenital anomalies, 76.5 (95% CI 58.9 to 99.4) for nervous system, 63.7 (95% CI 37.0 to 109.7) for digestive system, 55.3 (95% CI 42.5 to 72.1) for respiratory system, 32.1 (95% CI 17.8 to 57.9) for endocrine and 14.8 (95% CI 8.9 to 24.5) for circulatory system. External causes accounted for 46% of control deaths, but the SMR for external-related deaths was still higher (3.6 (95% CI 2.2 to 5.8)) for pupils with intellectual disabilities. Deaths amenable to good care were common.
**Conclusion** Pupils with intellectual disabilities were much more likely to die than their peers, and had a different pattern of causes, including amenable deaths across a wide range of disease categories. Improvements are

## Strengths and limitations of this study

► Novel use of education records and record linkage to death records to study mortality in an unselected cohort of children and young people with intellectual disabilities.
► Due to the use of a whole country population, these results are well-powered and generalisable.
► Despite comprising a whole country population, our study was not large enough to delineate cause-specific mortality ratios by sex.
► This study was limited by lack of demographic and clinical diagnostic information, including the severity or cause of intellectual disabilities.
► Reliance on death certificate data is limited by inconsistencies in reporting of cause of death

needed to reduce amenable deaths, for example, epilepsy-related and dysphagia, and to support families of children with life-limiting conditions.

## INTRODUCTION

Children and young people with intellectual disabilities have a much higher prevalence of physical and mental ill-health compared with the general population.[1–3] The life expectancy of people with intellectual disabilities has been reported to be about 20 years shorter than in the general population, or 28 years shorter specifically for people with Down syndrome.[4–7] While the actual number of deaths in childhood is smaller than in adults, mortality studies comparing people with intellectual disabilities with the general population have tended to show increased risk ratios in younger age groups compared with adults. However, the reported excess risk varies considerably between studies, and not all studies are comparable due to, for example, reporting deaths within different

age ranges, and additionally, some have small sample sizes and wide CIs. Reported standardised mortality ratios (SMRs) comparing people with and without intellectual disabilities, have ranged from 3.3 (95% CI 2.1 to 5.0) in young people aged 10 to 19 years[8] to 17.3 (95% CI 9.4 to 29.0) in young people aged 10 to 17 years[9]; from 2.6 in males aged 2 to 19 years and 1.7 in females aged 2 to 19 years,[10] to 21.6 (95% CI 10.8 to 38.7) in males aged 0 to 19 years and 18.1 (95% CI 3.7 to 53.0) in females aged 0 to 19 years;[11] and have been reported to be 30.4 (95% CI 18.4 to 47.5) in children aged 0 to 9 years.[9] We have summarised all previous studies to our knowledge which report mortality ratios for children and young people under age 25 years, with and without intellectual disabilities, where they are reported separately from older age groups (online supplementary appendix 1).

Most of these studies do not report causes of death among children and young people with intellectual disabilities. Bourke et al[12] reported the most common causes of death in children, young people and adults with intellectual disabilities aged 1 to 25 years to be respiratory infection (34%), with an additional 10% having an aspiration-related cause, congenital heart defects (15%) and accidents (11%). Compared with children and young people who did not have intellectual disabilities, their causes of death by International Classification of Diseases, tenth revision, (ICD-10) chapter were more likely to be attributed to the nervous system, endocrine, nutritional and metabolic diseases, or congenital malformations, and less likely to be attributed to conditions originating in the perinatal period, external causes, or injury or poisoning.[12] Patja et al[10] reported respiratory diseases to be the most common underlying/immediate cause of death in children and young people with intellectual disabilities aged 2 to 19 years, with a relative risk of 5.8 (95% CI 4.4 to 15.6) in males and 4.3 (95% CI 0.3 to 4.7) in females, and did not find any other causes (infectious diseases, tumours, vascular diseases, diseases of digestive system, accidents and poisonings, or other causes) to differ from those expected in the general population. However, the study was limited by small sample size. Durvasula et al[13] reported 7 of 14 deaths among young people with intellectual disabilities aged 10 to 24 years were attributed to the respiratory system (pneumonia and aspiration).

Adults with intellectual disabilities are over-represented in deaths which would have been amenable to treatment by timely and effective healthcare.[4 5 9] However, there is limited evidence on whether children and young people with intellectual disabilities also experience such amenable deaths more commonly than other children and young people, as most authors who have reported cause-specific mortality did so by grouping across all ages due to sample sizes.

Overall, as shown in online supplementary appendix 1, studies on mortality in children with intellectual disabilities are mostly small in size, and results are variable. Studies of causes of death exclusively in children and young people with intellectual disabilities are also limited. Hence, the aim of this cohort study is to compare all-cause and cause-specific mortality in Scotland's school attending population with and without intellectual disabilities.

## METHODS

We used education data from Scotland's annual pupil census between 2008 and 2013 to establish a cohort of children and young people with and without intellectual disabilities. We used individual record linkage based on probabilistic record matching (on date of birth, sex and postcode) to the Community Health Index, Scotland's list of all unique patient identifiers, including the National Records of Scotland (NRS) deaths register, to ascertain all deaths up to February 2015 in Scotland.

The Scottish annual pupil census is completed in September each year and provides information on all children attending local authority-funded primary, secondary and special schools in Scotland, or funded placements in alternative schools, which includes 95% of the entire population of children and young people in Scotland. This information includes whether the child has a record of Additional Support Needs, and the type of Additional Support Need. It is held by the Scottish Exchange of Educational Data (ScotXed).

The record linkage methodology required date of birth, sex and postcode; however, since names were not used to link pupil records to the health data, we excluded non-singleton births (available for Scottish-born pupils only, identified from linkage to maternity records). Unlikely matches were excluded and the most likely match was selected as the correctly linked pupil record. We also excluded any records with duplicate pupil records or where the linkage was tied with another patient. We included in the study all pupils with records of Additional Support Need due to intellectual disabilities between 2008 and 2013, between the ages 4 and 19 years old, on entry. Pupils were also censored on reaching age 25, if they reached this age during the observation period; so the maximum follow-up age was 24 years. Only pupils with intellectual disabilities recorded in at least two different school years were included in the intellectual disabilities group, to ascertain that they were correctly identified. Pupils who were included in at least two pupil census over the study period and had no record of intellectual disabilities or autism were used as the comparison group. Pupils with only autism were also excluded from controls, to eliminate potential mislabelling of support need for either autism or learning disability in the absence of clinical diagnoses.

The pupil census also includes data on age, sex, ethnicity and Scottish Index of Multiple Deprivation 2012 (SIMD).[14] Derived from individual pupil postcode of residence, SIMD is a composite of seven indices to indicate the extent of neighbourhood deprivation. SIMD was divided into quintiles according to the general population. Data on disability requirements including physical

(eg, visual, hearing or physical impairments), communication or curriculum needs are also listed.

Non-modifiable descriptive data on sex, ethnicity and SIMD, were taken from each pupils' first year in the census. For disability requirements, all records across multiple pupil census years were used to define whether having ever received adaptation requirements. Explorative statistical analyses using t-tests and $\chi^2$ tests were employed to investigate characteristics of pupils with intellectual disabilities compared with their peers in the comparison group. Differences in age of death were explored using t-tests. Crude mortality rates were calculated using the censor date 13 February 2015 or date of death. Since only those pupils who attended school in at least 2 years over our observed study period were eligible, the period between the first and second record introduced an immortal time bias, where no deaths could have occurred, and therefore the entry to the study was defined as the date of their second pupil census record. For indirect standardisation, observed deaths were assumed to be independent and vary with the Poisson distribution. The mortality rates were indirectly standardised for both males and females using the expected age-specific mortality rates per 1-year age group, using Stata's 'strate' command, to calculate age-SMRs for pupils with versus without intellectual disabilities. The 95% CIs were calculated based on the quadratic approximation of the log likelihood. Expected rates were calculated using fixed age and sex-specific rates from the large control population. The SMRs were subsequently calculated stratified by age into childhood (aged 5 to 14 years) and young people (aged ≥15 years), and by sex. The SMRs were also calculated for all deaths, excluding for external causes. This was to investigate whether the over-representation of female deaths in people with intellectual disabilities compared with the general population[12 15 16] is related to the large proportion of male deaths from external causes in the general population.[17]

For all-cause mortality, Kaplan-Meier survival curves were plotted for the overall time period for both groups. Cox proportional hazards models are also presented, adjusted for age and sex.

For cause of death analyses, the underlying cause of death is defined internationally[18] as the disease or injury which initiated the chain of morbid events leading directly to death, or the accident/act which produced the fatal injury. We also used a broader definition to analyse all-contributing causes, that included all deaths, with any mention on the death certificate related to the cause; combining both the underlying cause with secondary or contributing factors. While the same ICD-10 codes are used, it is important to note that one death may have several other additional causes as contributing factors, all of which are counted in figures reporting 'all-contributing causes'.

For the underlying causes of death, the total number of deaths in each ICD-10 chapter were collated, and this was then repeated for specific causes listed within chapters. Any errors or ambiguous deaths were listed as an unknown cause. All deaths where the underlying cause was ill-defined or defined by ICD-10 WHO guidelines[18] as codes in Chapter 18 excluding R95, were also re-classified as 'unknown'. Next, the breakdown of all-contributing causes were analysed by collating number of deaths in each ICD-10 chapter. For cause-specific SMRs, indirect age-standardisation was also performed, but using 5-year age bands to age-standardise rates and robust standard errors were used. For categories which had fewer than 10 deaths, no calculation was attempted due to lack of reliability in the small number of deaths. Furthermore, in keeping with the Office of National Statistics (ONS) mortality methodology,[19] all mortality rates between 10 and 20 deaths were labelled as unreliable. The ONS revised definition of avoidable mortality for children and young people[20] defined avoidable mortality as either amenable mortality (avoidable through good quality healthcare even after a condition has developed) or preventable mortality (avoidable through incidence reduction via public health interventions) or both. This list of ICD-10 causes was used to determine the occurrence of avoidable deaths. The rates and age-SMRs (age-standardised using 5-year age bands) for avoidable, amenable and preventable mortality were calculated using robust errors, except where there were fewer than 10 deaths per chapter. In keeping with the ONS avoidable mortality methodology,[19] all mortality rates based on fewer than 20 deaths were labelled as unreliable.

### Sensitivity analysis

A sensitivity analysis was carried out using wider inclusion criteria from the education data for both groups; the intellectual disabilities group included all pupils with at least one record of support at school due to intellectual disabilities. The control group included all pupils with at least one census record, and without support records for intellectual disabilities or autism. There were no other methodological changes made to the age standardising process or censor dates, but entry date was changed to the date of the first record of support need for pupils with intellectual disabilities or the first census date for pupils without intellectual disabilities.

All statistical analyses were undertaken using Stata, V.15.0 (StataCorp).

### Personal and patient involvement

This study was undertaken in the Scottish Learning Disabilities Observatory due to the growing concern among people with intellectual disabilities and their families around mortality. Its steering group includes people with intellectual disabilities and partners from third sector organisations. Results from this study will be disseminated to people with intellectual disabilities and their families in an easy-read version via the Scottish Learning Disabilities Observatory website and newsletters.

**Table 1** Demographic information for pupils with and without intellectual disabilities

| Demographic information* | Intellectual disabilities | | Controls | | P value† |
|---|---|---|---|---|---|
| Total, n (person-years) | 18 278 | (67 342) | 777 912 | (3 672 224) | |
| Male sex, n (%) | 11 891 | (65%) | 389 160 | (50%) | p<0.001 |
| Age, person-years (%) | | | | | |
| <10 | 12 518 | (19%) | 995 297 | (27%) | |
| 10 to 14 | 28 297 | (42%) | 1 332 123 | (36%) | |
| 15 to 19 | 23 672 | (35%) | 1 178 608 | (32%) | |
| 19 to 24 | 2855 | (4%) | 166 196 | (5%) | |
| Disability adaptations, n (%) | | | | | |
| Physical adaptation, ever received | 1971 | (11%) | 1837 | (0.2%) | p<0.001 |
| Curriculum adaptation, ever received | 6623 | (36%) | 6341 | (0.8%) | p<0.001 |
| Communication adaptation, ever received | 3553 | (19%) | 1760 | (0.2%) | p<0.001 |
| SIMD quintile, n (%) at first census | | | | | |
| 1 (most deprived) | 5822 | (32%) | 169 038 | (22%) | |
| 2 | 3888 | (21%) | 149 290 | (19%) | |
| 3 | 3397 | (19%) | 152 415 | (20%) | |
| 4 | 2896 | (16%) | 158 228 | (20%) | |
| 5 (least deprived) | 2275 | (12%) | 148 941 | (19%) | p<0.001 |
| Ethnicity, n (%) | | | | | |
| White‡ | 16 553 | (91%) | 708 941 | (91%) | p<0.001 |
| Asian‡ | 514 | (3%) | 23 791 | (3%) | |
| Mixed or multiple ethnicities | 144 | (1%) | 8035 | (1%) | |
| African, Caribbean or black | 87 | (<1%) | 4710 | (<1%) | |
| Other ethnic groups | 92 | (<1%) | 4665 | (<1%) | |
| Not disclosed / or unknown | 888 | (5%) | 27 770 | (4%) | |

*Data taken from first census record, except for disability adaptation, which includes any record across census years.
†$\chi^2$ test for intellectual disabilities compared with control group (For SIMD, $\chi^2$ test was performed across all categories, overall p value).
‡(White: Scottish, British and other) (Asian: Indian/British/Scottish, Pakistani/British/Scottish, Bangladeshi/British/Scottish, and Chinese/British/Scottish).
SIMD, Scottish Index of Multiple Deprivation.

## RESULTS

Out of 947 922 pupils in the census between 2008 and 2013 who were successfully linked to health records, there were 27 140 pupils who had ever registered as having an Additional Support Needs due to intellectual disabilities, and of these, 18 278 (1.9% of pupils) met the criteria of having at least two records of support. The remaining 8862 pupils with a single support record were excluded, except for the sensitivity analysis. There were 909 688 pupils without any records of intellectual disabilities or autism. Of these, 131 776 were excluded due to appearing in only 1 year of the census, except for the sensitivity analysis. The remaining 777 912 pupils attended school for at least 2 years over the study period and were designated as controls.

Using data from the pupils' first year in the census, pupils with intellectual disabilities were more likely to be male, more likely to reside in areas of greater neighbourhood deprivation and to have registered for free school meals, compared with their peers (table 1). Pupils with intellectual disabilities were also more likely to require adaptations in school, including physical adaptations, communication and curriculum adaptations. The majority of the study population were identified as having white (Scottish, British or other) ethnicity.

### Missing education support records

There were 11 329 pupils (62%) of the intellectual disabilities group who appeared in certain census years without having a record of support. The majority, 70%, (n=7970) were before the accrual of the first record; these pupils had a median two pupil census records prior to receiving their support (IQR 1, 3). There were 3359 pupils or 18% of the entire study group who went on to have census records without support records, after having received intellectual disabilities support provision. These pupils had a median 1 subsequent year (IQR 1, 2) without support out of a median 4 remaining years (IQR 3, 6) in the census.

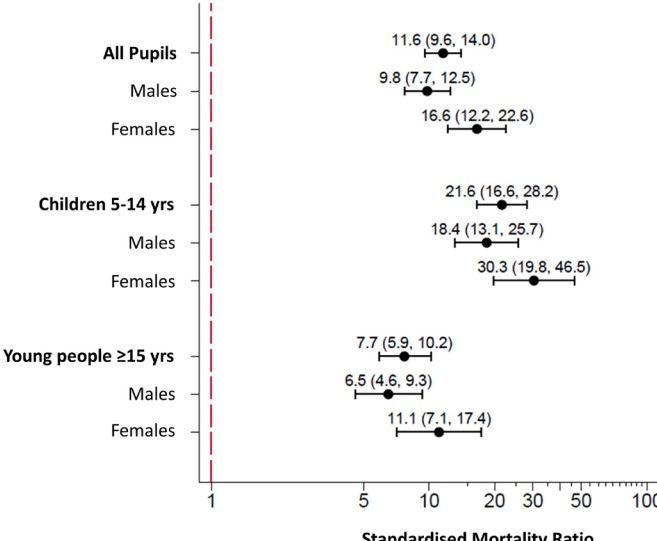

**Figure 1** Forest plot of age-standardised and sex-specific mortality ratios for pupils with intellectual disabilities.

## Mortality analysis

Linking the pupil census population to the NRS register of deaths up to February 2015 resulted in the equivalent of 3 739 568 person-years of follow-up. There were 564 deaths identified in the study population during this period. There were 106 deaths (0.6%) among children and young people with intellectual disabilities over 67 342 person-years, which translated to a crude mortality rate of 157 deaths per 100 000 person-years (95% CI 130 to 190). In the control group, there were 458 deaths (<0.1%) over 3 672 224 person-years, which translated to a crude mortality rate of 12 deaths per 100 000 person-years (95% CI 11 to 14). The mean age of death among children and young people with intellectual disabilities was 14.3 years (95% CI 13.4 to 15.1), which was significantly lower (p<0.001) than controls where the mean age of death was 16.1 years (95% CI 15.8 to 16.5). Sixty-two per cent of deaths among children with intellectual disabilities occurred in males, which was equivalent to the sex distribution in the whole intellectual disabilities cohort (p=0.545). Among controls, 61% of deaths occurred in males in spite of them accounting for only 50% of this group (p<0.001). Over 50% of deaths among pupils with intellectual disabilities occurred during childhood (<15 years old), compared with 29% of deaths among controls.

The all-cause age-SMR was 11.6 (95% CI 9.6 to 14.0), as shown in figure 1. The SMR was higher for female pupils than male pupils with intellectual disabilities; female SMR of 16.6 (95% CI 12.2 to 22.6) versus male SMR of 9.8 (95% CI 7.7 to 12.5). Exclusion of external causes of death resulted in a considerable increase in the all-cause SMR for both females and males with intellectual disabilities; overall SMR was 21.6 (95% CI 17.8 to 26.3), female SMR of 25.6 (95% CI 18.8 to 34.9) versus male SMR of 19.6 (95% CI 15.3 to 25.2). This produced a relative increase of 10 more deaths overall for pupils with versus without intellectual disabilities, which was similar

in females (+9.0 increase) and males (+9.8 increase). The childhood (aged 5 to 14 years) SMR was 21.6 (95% CI 16.6 to 28.2) and was higher for females than males with intellectual disabilities; female SMR of 30.3 (95% CI 19.8 to 46.5) versus male SMR of 18.4 (95% CI 13.1 to 25.7). For young people (≥15 years old), SMR was 7.7 (95% CI 5.9 to 10.2) and was also higher for females than males with intellectual disabilities; female SMR of 11.1 (95% CI 7.1 to 17.4) versus male SMR of 6.5 (95% CI 4.6 to 9.3). Hence, the difference from the control pupils was greater in children rather than young people for both females and males.

The Cox proportional HR for all-cause mortality, adjusted for age and sex, was found to be very similar; HR: 11.97 (95% CI 9.64 to 14.86). Proportional hazards assumption was met (p=0.422). Kaplan-Meier survival curves for the overall time period are found in online supplementary data (online supplementary appendix 2).

## Cause of death

Cause of death data was available for over 95% of deaths among pupils with intellectual disabilities and over 91% deaths among controls. Table 2 shows the underlying causes of death and all-contributing causes of death by ICD-10 chapter. There were major differences between pupils with intellectual disabilities and controls with regard to the most common underlying causes. Among pupils with intellectual disabilities, these were diseases of the nervous system (33%), congenital malformations, deformations and chromosomal abnormalities (22%), followed by nutritional, metabolic and endocrine diseases (8%), of which, most were conditions which were the cause of the pupils' intellectual disabilities, for example, neuronal ceroid lipofuscinosis or ornithine metabolism disorders. These were followed by respiratory diseases (8%) and neoplasms (7%). The most common underlying cause of death among control pupils was death due to external causes (46%), which made up a higher proportion of all deaths than in the pupils with intellectual disabilities (5%). Among controls, 71% of deaths due to external causes occurred in boys compared with 100% in the intellectual disabilities group.

There were also differences in the most common all-contributing causes of death (table 2). These chapters were not mutually exclusive, since one death could be included in several categories. Of the 106 deaths among pupils with intellectual disabilities, diseases of the nervous system contributed to 56 and diseases of the respiratory system contributed to 55. The 56 deaths which included diseases of the nervous system included 34 due to cerebral palsy and 16 due to epilepsy. The 55 deaths which included diseases of the respiratory system included 27 due to pneumonia, 9 due to pneumonitis associated with food and gastric contents, 17 due to respiratory failure, and 15 other respiratory disorders. In comparison, the control pupils had diseases of the nervous system contributing to 39 out of the total 458 deaths, and diseases of the respiratory system contributing to 51 of 458 deaths which

**Table 2**  By ICD-10 chapter, the underlying causes of death as well as all-contributing factors in death, and cause-specific crude mortality rates per 100000 person-years for pupils with and without intellectual disabilities

| ICD-10 chapter* | Underlying cause of death | | | | | | All-contributing factors in death | | | | | |
| | Intellectual disabilities | | | Controls | | | Intellectual disabilities | | | Controls | | |
| | n (%) | CMR | 95% CI | n (%) | CMR | 95% CI | n | CMR | 95% CI | n | CMR | 95% CI |
|---|---|---|---|---|---|---|---|---|---|---|---|---|
| Ch. 6. Diseases of the nervous system | 35 (33%) | 51.9 | 37.3 to 72.4 | 19 (4%) | 0.5U | 0.3 to 0.8 | 56 | 83.2 | 64.0 to 108.1 | 39 | 1.1 | 0.8 to 1.5 |
| Ch. 17. Congenital malformations, deformations and chromosomal abnormalities | 23 (22%) | 34.2 | 22.7 to 51.4 | 13 (3%) | 0.4U | 0.2 to 0.6 | 32 | 47.5 | 33.6 to 67.2 | 18 | 0.5U | 0.3 to 0.8 |
| Ch. 4. Endocrine, nutritional and metabolic diseases | 9 (8%) | | | 15 (3%) | 0.4U | 0.2 to 0.7 | 11 | 16.3U | 9.0 to 29.5 | 18 | 0.5U | 0.3 to 0.8 |
| Ch. 10. Diseases of the respiratory system | 8 (8%) | | | 17 (4%) | 0.5U | 0.3 to 0.7 | 55 | 81.7 | 62.7 to 106.4 | 51 | 1.4 | 1.1 to 1.8 |
| Ch. 2. Neoplasms | 7 (7%) | | | 92 (20%) | 2.5 | 2.0 to 3.1 | 8 | NA | | 94 | 2.6 | 2.1 to 3.1 |
| Ch. 20. External causes of morbidity and mortality | 5 (5%) | | | 210 (46%) | 5.7 | 5.0 to 6.6 | 16 | 23.8U | 14.6 to 38.8 | 231 | 6.3 | 5.5 to 7.2 |
| Ch. 9. Diseases of the circulatory system | <5 (5%) | | | 24 (5%) | 0.7 | 0.4 to 1.0 | 15 | 22.3U | 13.4 to 37.0 | 54 | 1.5 | 1.1 to 1.9 |
| Ch. 11. Diseases of the digestive system | <5 (5%) | | | 6 (1%) | | | 13 | 19.3U | 11.2 to 33.3 | 11 | 0.3U | 0.2 to 0.5 |
| Ch. 1. Certain infectious and parasite diseases | <5 (5%) | | | 12 (3%) | 0.3 | 0.2 to 0.6 | 8 | | | 29 | 0.8 | 0.5 to 1.1 |
| Ch. 5. Mental and behavioural disorders | <5 (5%) | | | <5 (1%) | | | 6 | | | 17 | 0.5U | 0.3 to 0.7 |
| Ch. 3. Diseases of the blood, blood-forming organs and immune mechanism | <5 (5%) | | | <5 (1%) | | | <5 | | | 8 | | |
| Ch. 14. Diseases of the genitourinary system | <5 (5%) | | | <5 (1%) | | | <5 | | | <5 | | |
| Ch. 13. Diseases of the musculoskeletal system and connective tissue | 0 | | | <5 (1%) | | | <5 | | | 7 | | |
| Ch. 8. Diseases of the ear and mastoid process | <5 (5%) | | | 0 | | | <5 | | | 0 | | |
| Ch. 15. Pregnancy, childbirth and puerperium | 0 | | | <5 (1%) | | | 0 | | | <5 | | |
| Ch. 18. Symptoms, signs and abnormal clinical and laboratory findings | NA | | | NA | | | 28 | | | 76 | | |
| Ch. 19. Injury, poisoning and certain other consequences of external causes | NA | | | NA | | | 10 | | | 219 | | |
| Unknown cause or error in underlying code | 5 (5%) | | | 39 (9%) | | | NA | | | NA | | |
| TOTAL | 106 | | | 458 | | | NA | | | NA | | |

U rates based on 10 to 20 deaths labelled 'U' for unreliable.

*n <5 repressed due to statistical disclosure.

CMR, crude mortality rate – reported for ≥10 deaths; ICD-10, International Classification of Diseases, tenth revision.

**Table 3** The top 10 specific underlying causes of death and all-contributing causes of death for pupils with and without intellectual disabilities

| Intellectual disabilities | | | | Controls | | | |
| --- | --- | --- | --- | --- | --- | --- | --- |
| Underlying cause of death | n | All-contributing factors | n | Underlying cause of death | n | All-contributing factors | n |
| Cerebral palsy | 19 | Cerebral palsy | 34 | All neoplasms | 92 | Signs and symptoms: injury | 114 |
| Brain deformity | 9 | Pneumonia | 27 | Traffic accident | 76 | All neoplasms | 94 |
| All neoplasms | 7 | Respiratory failure | 17 | Self-harm | 54 | Traffic accidents | 76 |
| Muscular dystrophy | 6 | Epilepsy | 16 | Accidents, other | 41 | Self-harm | 54 |
| Epilepsy | 5 | Respiratory disorders | 15 | External, undetermined intent | 25 | Signs and symptoms: asphyxiation | 51 |
| Chromosomal abnormalities | 5 | Brain deformity | 12 | Asthma | 14 | Accident, other | 43 |
| Neuronal ceroid lipofuscinosis | <5 | Chromosomal abnormalities | 10 | Assault | 13 | Signs and symptoms: poisoning | 29 |
| Pneumonia, including influenza | <5 | Pneumonitis due to food and gastric contents | 9 | Infections | 12 | All infections | 29 |
| Congenital heart disease | <5 | All neoplasms | 8 | Epilepsy | 8 | External, undetermined intent | 26 |
| Accidents, other | <5 | All infections | 8 | Cystic fibrosis | 8 | Pneumonia | 21 |
| Unknown causes | 5 | Ill-defined or ambiguous death | 8 | Unknown causes | 39 | Ill-defined or ambiguous death | 58 |

included 21 due to pneumonia. The most common all-contributing causes of death for the control pupils were, as found for the underlying cause, external causes at 50% compared with 15% among pupils with intellectual disabilities.

Table 2 reports these data by presenting the cause-specific crude mortality rates by ICD-10 chapter for all pupils. As recommended by the ONS,[19] avoidable mortality rates based on low numbers are labelled as unreliable and marked 'U'.

The top 10 individual leading causes of death are shown in table 3. Among pupils with intellectual disabilities, the highest number of individual underlying cause of deaths were cerebral palsy (18%), followed by congenital brain deformities (8%) and neoplasms (7%). Where there were fewer than five individual deaths per cause, these causes were not reported due to statistical disclosure control. For the majority of deaths in pupils with intellectual disabilities, this was the case; 85% of specific causes could not be disclosed. Among control pupils, the highest number of individual underlying cause of deaths were neoplasms (20%), and road traffic accidents (17%). In relation to their peers, only three of the top 10 underlying causes of death among children with intellectual disabilities featured in the top 10 list for the controls—neoplasms (7% vs 20% of controls), epilepsy (5% vs 2% controls) and accidents (non-road traffic related,<5% vs 9% controls).

Cause-specific SMRs, indirectly standardised using 5-year age bands and robust errors, are shown in figure 2.

For underlying causes, this was only possible for the two largest categories (by ICD-10 chapters); congenital malformations, deformations and chromosomal abnormalities, and diseases of the nervous system. For the all-contributing causes, the age-SMR for seven chapters were calculated. For congenital malformations, deformations and chromosomal abnormalities, the SMR was $98.8^U$ (95% CI 69.9 to 139.7), and for diseases of the nervous system was 76.5 (95% CI 58.9 to 99.4). The ratios were also high for diseases of the digestive system at $63.7^U$ (95% CI 37.0 to 109.7); and for diseases of the respiratory system at 55.3 (95% CI 42.5 to 72.1). Despite external causes contributing to a larger proportion of deaths among the control group, the mortality rate was still higher in the intellectual disabilities group than in the controls; the crude rate was $23.8^U$ per 100 000 person-years, compared with $6.3^U$ per 100 000 for the controls for external cause of death (either as the underlying cause or as a contributing factor). This produced an SMR of $3.6^U$ (95% CI 2.2 to 5.8), demonstrating there is considerable over-representation in the intellectual disabilities group versus the controls.

**Avoidable mortality**

According to the UK ONS definition of avoidable mortality, (deaths which are amenable, preventable or both), 19% of deaths in the intellectual disabilities cohort were classed as avoidable; 15% of deaths were amenable to treatment and 6% were preventable. The majority of avoidable deaths (80%) were considered amenable to

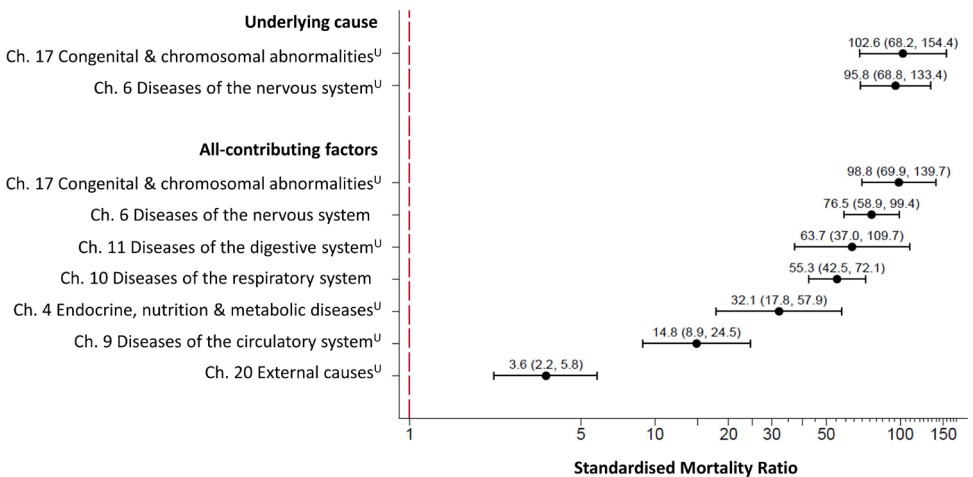

**Figure 2** Forest plot of cause-specific age-SMRs for pupils with intellectual disabilities by ICD-10 chapter for underlying cause of death and for all-contributing factors of death. Footnote: Age-SMRs and 95% CI were calculated using 5-year age bands for all ICD-10 chapters with ≥10 deaths. SMRs which were calculated using low numbers (between 10 and 20 deaths) are labelled 'U' for unreliable. ICD-10, International Classification of Diseases, tenth revision; SMR, standardised mortality ratio.

treatment for their age group, including epilepsy, pneumonia and neoplasms. Among the control pupils, 63% of deaths were classed as avoidable, 16% were amenable to treatment and 48% were preventable. The crude avoidable mortality rate for pupils with intellectual disabilities was higher at 29.7[U] (95% CI 19.2 to 46.0) per 100 000 in pupils with intellectual disabilities, compared with 7.8 (95% CI 7.0 to 8.8) per 100 000 in the control pupils. The SMR was 3.6[U] (95% CI 2.3 to 5.5). Further breakdown of avoidable rates was possible for deaths that were amenable to healthcare; in the intellectual disabilities group, the amenable mortality rate was 23.8[U] (95% CI 14.6 to 38.8) per 100 000 versus 2.0[U] (95% CI 1.6 to 2.5) per 100 000 in controls; and the SMR was found to be 11.5[U] (95% CI 7.0 to 18.8).

Among pupils with intellectual disabilities, there were additional causes of death that the authors of this paper consider would have been amenable to healthcare: aspiration pneumonia; otitis media; megacolon; gastrointestinal haemorrhage; gastroenteritis; and contributing causes of death including gastro-oesophageal reflux and urinary tract infections. These are not currently included within the ONS list of underlying causes.

### Sensitivity analysis

Of 27 140 pupils with at least one record of support due to intellectual disabilities, 65% were male, and compared with the main analysis group, there were significant reductions in frequency of school adaptations (physical disability reduced from 11% vs 9%, (p<0.001), curriculum adaptations from 36% to 31% (p<0.001) and communication adaptations from 19% to 16% (p<0.001)). There were higher numbers of pupils in this group with years without intellectual disabilities support. There were 156 deaths in the intellectual disabilities group (134 per 100 000 person-years (95% CI 114.2 to 156.3)) compared with 684 deaths (13.8 per 100,000 (95% CI 12.8 to 14.8)) among the control group. The SMR for this sensitivity

analysis was 9.5 (95% CI 8.1 to 11.1), a change of −2 excessive deaths compared with the main analysis SMR. Mean age of death was similar in the sensitivity group, being 14.4 years (95% CI 13.7 to 15.1) in the intellectual disabilities group and 16.2 years (95% CI 15.9 to 16.5) in the control group. The ratio of deaths by sex were also very similar, with no difference for the intellectual disabilities group; 61% deaths were in males, similar to the proportion of males in the group (p=0.306) and an increase in male deaths among controls; 63% deaths were in males, whereas only 50% in the control group were male (p<0.001).

### DISCUSSION

#### Principle findings and interpretation

Our study is one of very few that has reported mortality rates among children and young people with intellectual disabilities, and is highly novel in reporting underlying, all-contributing and the most common individual causes of death at this age, including cause-specific SMRs. We have demonstrated that children and young people in Scotland with intellectual disabilities have a 12-fold risk of death compared with their peers, rising to 22-fold on excluding external causes. Pupils with intellectual disabilities were also over-represented in deaths that were amenable to healthcare, and were approximately 3.6 times more likely to experience an avoidable death (although calculated using unreliably low rates). Children aged 5 to 14 years with intellectual disabilities had a higher risk relative to peers (SMR 21.6) than the young people aged ≥15 years with intellectual disabilities (SMR 7.7). This difference reflects that, in the general population, there were considerably more deaths in young people than in children, especially for males, as opposed to more deaths of children than young people with intellectual disabilities. The SMR was higher for female pupils in both age groups,

reflecting the higher death rate of males in the controls. Nervous system and respiratory causes of death were most common among children and young people with intellectual disabilities, including deaths that would have been amenable to quality healthcare, such as epilepsy, pneumonia and pneumonitis due to food and gastric contents. It is highly important to identify amenable deaths so that actions can be devised and taken. Causes of death among children and young people with intellectual disabilities were higher across several disease categories than for other children and young people, including diseases of the nervous system, digestive system, respiratory system, endocrine, nutrition and metabolic diseases, diseases of the circulatory system and external causes.

Previous studies have demonstrated that there is an increased risk of sudden unexpected death in epilepsy among people with intellectual disabilities; however, in our study, the majority of deaths which listed epilepsy as a contributing factor, also listed pneumonia, so this does not appear to account for our findings.

While external causes of deaths accounted for the greatest proportion of deaths among control children and young people (46%), especially in males, we found that external causes of death were still over-represented among children and young people with intellectual disabilities compared with their controls (partly due to inhalation of gastric contents and inhalation of objects obstructing breathing). Trollor et al[17] hypothesised that higher SMRs in adult women than men with intellectual disabilities may be driven by the larger proportion of male deaths in the general population due to external causes and the lack of equivalent deaths in males with intellectual disabilities. However, in our population of children and young people, when we re-calculated SMRs to exclude external causes, the observed increase in risk for females remained. Hence, at this age range, this is only a partial explanation for the sex differences in SMRs, and there are other risk factors and vulnerabilities which require further exploration. It should be noted, however, that in children and young people with intellectual disabilities, not all studies report a higher SMR in females compared with males.[10 11]

### Comparison with previous studies

Two previous studies[9 12] have reported a higher SMR for children than for young people. Glover et al[9] reported results separately for children aged 0 to 9 years (SMR 30.4) and young people aged 10 to 17 years with intellectual disabilities (SMR 17.3). The Australian study by Bourke et al[12] reported a higher adjusted HR (aHR) for children aged 6 to 10 years (aHR 12.6) than young people aged 11 to 25 years (aHR 4.9). The SMRs we report are lower than those reported by Glover et al,[9] but the extent of difference between the children and young people is similar, although for differently defined age groups. The CIs reported in our study are narrower due to the larger sample size. The SMRs we report are higher than those previously reported from small scale studies in Finland

and USA,[8 10 15] and a larger one in Ireland,[16] yet lower than a study reported from England[9] and a small study from Canada.[11] These differences may be due to actual international differences or due to methodological differences between studies including: the method and source of identification of the population with intellectual disabilities; age ranges included; and study size with several of the previous studies having produced results with wide CIs. All of these studies report a higher SMR in females than in males, except the studies conducted in Canada and Finland.

The only previous study that has reported cause of death for children and young people aged 1 to 25 years reported the most common causes of death to be infections in 50% (particularly respiratory infections in 34%), birth defects in 19% (particularly cardiac defect in 15%) and accidents in 11%, although by ICD-10 chapter, deaths due to diseases of the respiratory tract were reported for 4.6%, infections and parasitic diseases for 3.1% and external causes for 7.7%; and the most common were congenital malformations, deformations and chromosomal abnormalities in 29.1%, and diseases of the nervous system in 27.6%.[12] They did not report cause-specific SMRs by ICD-10 chapters, but crude numbers were proportionally higher for children with intellectual disabilities for diseases of the nervous system, endocrine, nutritional and metabolic diseases, and congenital malformations, and lower for conditions originating in the perinatal period, external causes, or injury or poisoning.[12] We demonstrated diseases of the nervous system and respiratory system to be the most common causes of death, and that cause-specific SMRs were higher across all congenital malformations, deformations and chromosomal abnormalities, diseases of the nervous system, digestive system, respiratory system, endocrine, nutrition and metabolic diseases, circulatory system and external causes.

Glover et al graphed avoidable deaths in his study of children and adults.[9] We are unaware of any previous studies numerically quantifying amenable deaths among children and young people with intellectual disabilities.

### Strengths and limitations

Our study drew on data from an entire country, collected annually, and linked to national death records. It was large in scale, including over 18 000 children and young people with intellectual disabilities and a large control population. A record of intellectual disabilities at school brings an entitlement to additional support and so is likely to drive good recording in high-income countries like Scotland. However, it only uses a binary definition for intellectual disabilities; therefore, the study could not investigate mortality among people with different causes and severities of intellectual disabilities. Our study was not large enough to delineate cause-specific mortality ratios by sex nor to study whether there are any ethnic variations. Use of death certificate data is known to have limitations,[19] including inconsistent reporting and no reporting of severity of conditions. There may be some

diagnostic overshadowing in death certificate data for people with intellectual disabilities, obscuring the events leading to death.[21–23] The ONS list of avoidable deaths does not include some that appear important among children and young people with intellectual disabilities, such as aspiration pneumonia, otitis media, megacolon, gastrointestinal haemorrhage and gastroenteritis, which featured as an underlying cause of death in our data. Additionally, death certificate data does not include wider determinants of health and death that may be implicated, such as being the target of discrimination or neglect.

Additionally, while we believe this population to be highly representative of children with intellectual disabilities across Scotland, we acknowledge that we were unable to access data on children not in school; there may be some under-ascertainment of children with intellectual disabilities with exceptional and complex health needs unable to attend school.

### Conclusion and future directions

It is extremely important to study deaths among children and young people with intellectual disabilities, especially as so few studies have previously done so. Among the studies that have, there exists wide variation in the extent of reported inequality compared with other children and young people, and wide CIs, but all show a higher SMR. Our large study provides robust data that quantifies the extent of the difference; children and young people have a 12 times higher risk of death. A larger body of research exists for adults (rather than children and young people) with intellectual disabilities, and demonstrates substantial inequalities and a high proportion of amenable deaths that could be addressed via reasonable adjustments in care provision. In our study, we have now reported that children and young people with intellectual disabilities also experience inequalities and experience amenable deaths. This is important, and we need a better understanding of it so that targeted improvements in care can be developed and delivered to reduce this inequality. Heslop et al[24] conducted a confidential inquiry into deaths of people with intellectual disabilities and made recommendations for improvements to practice regarding respiratory deaths, including aggressive monitoring and treatment of gastro-oesophageal reflux as well as postural and physical therapies. We have found that this is also important for children and young people with intellectual disabilities, if we are serious about improving life expectancy. Additionally, Scotland now offers influenza vaccines to all primary school-age children to reduce pneumonia; we therefore need to understand uptake by children with intellectual disabilities, and its determinants, to gauge whether this will change mortality findings.

The results of this study should be used to inform and direct multidisciplinary healthcare teams, as well as educators and carers to the associated risks of mortality in childhood and generate greater awareness around potential areas of improvement. Our countrywide study had a mean follow-up of around 5 years, and given that the pupil census is recorded annually, it presents the framework for further work to investigate both mortality trends in children and young people with intellectual disabilities, and a more detailed understanding of these. Future studies could consider looking at predictors of death in children and young people to inform translation of findings into clinical benefit for people with intellectual disabilities.

**Acknowledgements** We thank the Scottish Exchange of Educational Data (ScotXed) Pupil Census, the National Records of Scotland and Information Services Division (ISD) of National Health Service Scotland for providing data and the eData Research and Innovation Service team at ISD Scotland for assisting with the data linkage and analysis stage of this project.

**Contributors** GSS analysed the data, interpreted findings and wrote the first draft of the manuscript. MF developed record linkage, analysed the data, interpreted findings and contributed to the manuscript. JPP developed record linkage, interpreted findings and contributed to the manuscript. DK, AH and CM interpreted data and contributed to the manuscript. S-AC conceived the study, analysed and interpreted the data and contributed to the manuscript. All authors approved the final version of the manuscript.

**Funding** This study was funded by an MRC Mental Health Data Pathfinder Award (MC_PC_17217), and the Scottish Government via the Scottish Learning Disabilities Observatory.

**Disclaimer** The funders had no role in the study design, collection, analyses or interpretation of data, writing the report nor the decision to submit the article for publication.

**Competing interests** None declared.

**Patient and public involvement** Patients and/or the public were involved in the design, or conduct, or reporting, or dissemination plans of this research. Refer to the Methods section for further details.

**Patient consent for publication** Not required.

**Ethics approval** This study received approval from the NHS National Services Scotland Privacy Advisory Committee and Public Benefit and Privacy Panel -(PBPP) approval no. 1617-0259.

**Provenance and peer review** Not commissioned; externally peer reviewed.

**Data availability statement** No data are available. This study linked patient information held across several administrative health datasets within Information Services Division (ISD) of NHS National Services Scotland (NSS), with externally held data held by the Scottish Government (ScotXed education) and National Records of Scotland. Linkage and de-identification of data was performed by ISD. A data processing agreement between NHS NSS and University of Glasgow and a data-sharing agreement between ScotXed and University of Glasgow were drafted. The University of Glasgow were authorised to receive record-linked data controlled and held by ISD within NSS, via access through the national safe haven. The ISD Statistical Disclosure Control Protocol was followed. It is therefore not possible to share data with other parties.

**ORCID iDs**
Gillian S Smith http://orcid.org/0000-0001-7241-6951
Angela Henderson http://orcid.org/0000-0002-6146-3477
Sally-Ann Cooper http://orcid.org/0000-0001-6054-7700

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
