## [Reviewer comments · BMJ Open]

ARTICLE DETAILS

TITLE (PROVISIONAL)	Rates and causes of mortality among children and young people with and without intellectual disabilities in Scotland: a record linkage cohort study of 796,190 schoolchildren
AUTHORS	Smith, Gillian; Fleming, Michael; Kinnear, Deborah; Henderson, Angela; Pell, J. P.; Melville, Craig; Cooper, Sally-Ann

VERSION 1 – REVIEW

REVIEWER	Julian Trollor University of New South Wales, Department of Developmental Disability Neuropsychiatry
REVIEW RETURNED	26-Sep-2019

GENERAL COMMENTS	Thank you for asking me to review this manuscript which presents a largescale study of mortality in people with Intellectual disability compared to children and younger people without intellectual disability. The study population is taken from schools data. She key strengths are the large datasets, and comprehensive analysis including age-specific standardised mortality rates, and all-cause mortality data. It is also pleasing to see consideration of the contribution of external causes of mortality to the age and gender specific differences between ID and non-ID groups. Overall, the study presents new findings, is well described, the statistical methods appropriate, the results clearly written, and the discussion appropriate to the topic. Title and Abstract The title could be misconstrued as 800,457 children with intellectual disability. The term Age-SMR appears in the abstract but has not been defined. “External causes accounted for 46% of control deaths, but SMR was still higher (3.5 {2.2, 5.8}) for pupils with intellectual disabilities.” It should be clarified whether the SMRs being referred to relate specifically to external causes of death in both groups. Last sentence of the abstract is overly broad and could possibly be edited, together with the second last sentence of the conclusion into a more specific sentence relating to translating the findings into action. Introduction Page 3, sentence beginning with “reported standardised mortality ratio” lacks specificity- it should be clarified that it refers to people with intellectual disability Page 5, “mortality studies comparing people with intellectual disabilities with the general population have shown increased risk ratios in younger age groups compared to adults”- as is
---

	subsequently acknowledged, this is not uniform across studies, so softer statement or more detail regarding the findings is advisable. Page 5, The end of the first paragraph of the introduction is a little hard to read and it may be better to refer the reader to the table. It should be clarified whether table one is meant to be inclusive of all studies, which report mortality in children and younger people with ID, or how the final list was arrived at- as at present there are some publications missing. The overall aim of the study “ the aim of this longitudinal cohort study is to compare all-cause and cause-specific mortality in Scotland’s school-aged population with and without intellectual disabilities” Is slightly different from actual data set which is derived from schools. it would be better framed as “.....Scotland's school attending population with and without intellectual disabilities” Methods Page 6: What is known about the makeup of the 5% of young people not captured in the Scottish annual pupil census? could it be that people with intellectual disability and complex needs, who may be at higher risk of death, and who have different causes of death, are more likely to be in the 5% that is not surveyed? Page 7: Why were non-singleton births excluded? this seems unnecessary and is unexplained. The database would appear to be able to distinguish between support to related to an autism specific issue or intellectual disability. it is therefore unclear why people with autism without intellectual disability were not included in the control sample. The concept of immortal time bias may be unfamiliar to some readers, as it was a new term to me. I was interested in why some of the data is presented as childhood (5-14) and young people 14+) and other data is presented as 5 year age bands. it might be useful to be consistent across the analyses. Results Page 11: the statement regarding the minimum number of deaths in particular categories that were reported seems to vary from 10 to 20. Later in the Results section of the manuscript, there is also reference to a minimum cell size of 5. Clearer disambiguation may be helpful for the reader as I was uncertain about these distinctions. Page 14: there may be value in documenting the change in SMR for people without intellectual disability once external causes of death were excluded? A comment on the relative magnitude of the change in males and females with intellectual disability once external causes of death were excluded might also be warranted, Though arguably, this could be left to the discussion. Page 15: the last part of the sentence at line 19 “there were also differences in the most common or contributing causes of death-those with any mention on the death certificate” is not necessary as this has been clearly stated in the methods section. Rather than by chapter number, Table 3 may be better organised according to most frequent causes of death in one or other of the cohorts. Discussion
--	--

	This is clearly expressed. Conclusions and future directions Future studies could consider looking at predictors of death in children and young people to inform translation of findings into clinical benefit for people with ID. Ethics Approvals It is usual to quote ethics committee approval numbers in my jurisdiction. Should this be included?
--	--

REVIEWER	Jenny Bourke Telethon Kids Institute Australia
REVIEW RETURNED	15-Oct-2019

GENERAL COMMENTS	This paper has addressed an important area of inequality through analysing the mortality rates of young people with intellectual disability (ID) compared with the general population. Using a cohort of students from census data, the study found 1.8% were identified with an intellectual disability, a prevalence very close to other estimates of population prevalence of ID, indicating there was appropriate identification of ID in the cohort. The study has used data linkage methodology to identify those who died over the study period. The findings reflect the much higher rates of mortality in those with ID, particularly for those <15 years of age and in females. The study has investigated cause of death and specifically looked at the effect of external cause of death on standardised mortality ratios. This is a cause representing almost half the underlying cause of deaths in the controls and yet the SMR was still 3.5 for those with ID. The study investigated whether the removal of external causes, generally related to males in the general population, may influence the increased mortality rates seen in females. However the removal of external causes did show an increase in SMR overall but also an increase for both males and females. This is helpful in partly explaining the higher mortality seen in females but leaves further investigation needed. The study has broadened the knowledge in the area of cause of death in children and young people with ID - and importantly identified those deaths amenable to high quality health care. Most importantly it has highlighted the inequality that exists for this vulnerable group and is a well-written and presented paper. There is only one minor typo noted- page 7, line 35 :except where there were (for) fewer than.
---

REVIEWER	Dr Sarah Lay-Flurrie University of Oxford, UK
REVIEW RETURNED	14-Nov-2019

GENERAL COMMENTS	This paper seeks to understand mortality risk in children with learning disabilities compared to other children, using administrative data from Scotland. Overall, this paper is clear and the study well conducted. I have the following specific comments/queries: Introduction: The introduction provides good background and justification for the study. As Table 1 prevents more detailed information from previous studies only, it may be more appropriate to include this in online appendix rather than the main body of the paper.
---

	Methods: The methods are generally clear and the authors have carefully designed the study to avoid immortal time bias. The data sources and statistical methods are also appropriate. Some more detail is required in the following areas:  1. The decision to include non-singleton births could be more clearly justified. 2. Although the authors state when follow-up started for each child, they do not state when it ended. As the authors were interested in deaths in children/ young people and they criticise previous work for inclusion of adult deaths, did they censor patients/ end follow-up at age 19? 3. The data linkage process is not entirely clear based on the information provided. Although the linkage was done by a third party by National Services Scotland the authors should still be able to report if linkage was based on exact or probabilistic matching, the variables on which linkage was carried out and how inexact matches were handled. 4. Chapters for ICD10 codes R00-R94 and R95-R99 are mentioned in the methods but then subsequently referred to as Chapters 18 and 19 in the results section - to aid the reader it would be preferable if the description was consistent. Results: The results are well presented and the figures are clear. The authors have been thorough in ensuring cell suppression where required. There is some repetition of results given in both the text and tables. The authors may wish to consider condensing some of this material and/or moving some results to an online appendix. Ethical approval: The authors should provide the review number/ approval number for the ethical approval received.
--	--

REVIEWER	Rebecca Bendayan University College London, Population, Policy and Practice Programme
REVIEW RETURNED	26-Nov-2019

GENERAL COMMENTS	This study aims to investigate mortality rates and causes in children and young people with intellectual disabilities using a retrospective cohort with record linkage. A major strength of this study is the data source as the authors used education data from Scotland's annual pupil census to identify individuals with and without intellectual disabilities; and its linkage to Scotland deaths registry, to account for all deaths up to February 2015. Overall, this work is of great interest and can be very useful to guide public health policy. Getting access, preparing and analysing data from linkages is challenging and this should be acknowledged. This review focuses on the methods and statistical approach and my overall comment is related to the need of more details in what has been done and why. In order to ensure replicability, we need the methods section to be more complete. From the methods section. I have some questions for the authors. While I totally understand the exclusion of non-singleton births, I wonder why only pupils with intellectual
--

	disabilities recorded in at least two different school years were included in the intellectual disabilities group. Does this mean that those that had only recorded intellectual disability in one school year were excluded from the whole study? I also understand that those that had only one-year record of no intellectual disability were also excluded. Was then an inclusion criterion to have at least two records over time to be part of the sample? If this is the case? What do we know about those that have only one record? I can understand this a strategy to ascertain that they are correctly identified as individuals of the intellectual disabilities group, but why not just run sensitivity analyses comparing those with 1 school year and those that were at least in two years? And same with comparison group? Can the authors please provide some numbers about how many were excluded for this reason and how might they differ from the sample that was considered in the study. This is particularly relevant to understand to what extent these results are generalizable. Were there any individuals that had two years or more with intellectual disabilities and two years or more without? If yes, how were these handled? About the follow up and the modelling approach: Education data was extracted between 2008 and 2013 (that is, over 5 years) and we understand that deaths were from 2008 to February 2015 in Scotland. Is this correct? Then there might be a lot of variability in length of follow up across individuals? If yes, how did the authors deal with this? I am missing more details on the statistical approach followed. More information on the modelling approach, how the models were built, whether robust approaches for smaller samples (in the case of mortality cause) were considered? Where the assumptions checked? Some survival graphs will also contribute to a better understanding of the results. All this information would be also useful to explore other limitations that might have not been mentioned. Minor: Table 2 refers to baseline data which we assume is 2008. It would help that is clear in the title of the table. In the abstract there is no sentence summarizing the statistical approach.
--	---

VERSION 1 – AUTHOR RESPONSE

Reviewer 1 Comments

Title / Abstract

1. The title could be misconstrued as 800,457 children with intellectual disability.

- *Response - We agree with the reviewer and have changed the title to: “Rates and causes of mortality among children and young people with and without intellectual disabilities in Scotland: a record linkage cohort study of 796,190 schoolchildren” to emphasise that the total study number includes our control population.

2. The term Age-SMR appears in the abstract but has not been defined.

- *Response - We have changed the abbreviation in the abstract to “standardised mortality ratios (SMRs)” to ensure this has been defined.

3. "External causes accounted for 46% of control deaths, but SMR was still higher (3.5 {2.2, 5.8}) for pupils with intellectual disabilities." It should be clarified whether the SMRs being referred to relate specifically to external causes of death in both groups.

- *Response - We agree and have changed this to: "External causes accounted for 46% of control deaths, but the SMR for external-related deaths was still higher (3.5 {2.2, 5.8}) for pupils with intellectual disabilities."

4. Last sentence of the abstract is overly broad and could possibly be edited, together with the second last sentence of the conclusion into a more specific sentence relating to translating the findings into action.

- *Response - We agree and have amended the abstract to: "Improvements are needed to reduce amenable deaths e.g. epilepsy-related and dysphagia, and to support families of children with life-limiting conditions."

Introduction

1. Page 3, sentence beginning with "reported standardised mortality ratio" lacks specificity- it should be clarified that it refers to people with intellectual disability

- *Response - We agree, and have changed the text to: "Reported standardised mortality ratios (SMR) comparing people with and without intellectual disabilities, have ranged from...."

2. Page 5, "mortality studies comparing people with intellectual disabilities with the general population have shown increased risk ratios in younger age groups compared to adults"- as is subsequently acknowledged, this is not uniform across studies, so softer statement or more detail regarding the findings is advisable.

- *Response - We agree, and have changed this to: "While the actual number of deaths in childhood is smaller than in adults, mortality studies comparing people with intellectual disabilities with the general population have tended to show have shown increased risk ratios in younger age groups compared to adults."

3. Page 5, The end of the first paragraph of the introduction is a little hard to read and it may be better to refer the reader to the table.

It should be clarified whether table one is meant to be inclusive of all studies, which report mortality in children and younger people with ID, or how the final list was arrived at- as at present there are some publications missing.

- *Response - We have changed this section of the paragraph: "We have summarised all previous studies to our knowledge which report mortality ratios for children and young people under aged 25, with and without intellectual disabilities, where they are reported separate from older age groups (Table S1 -Supplementary data)".

- We have also checked the literature search again and added age-specific rate ratios for any age group reported separately for people under age 25, including from supplementary data. An additional 3 studies have now been added to Table S.1.

4. The overall aim of the study " the aim of this longitudinal cohort study is to compare all-cause and cause-specific mortality in Scotland's school-aged population with and without intellectual disabilities" Is slightly different from actual data set which is derived from schools. it would be better framed as ".....Scotland's school attending population with and without intellectual disabilities"

- *Response - We agree and have altered the manuscript to "Hence, the aim of this cohort study is to compare all-cause and cause-specific mortality in Scotland's school attending population with and without intellectual disabilities".

- We have included age groups in Table 1. to clarify the age-range of the dataset.

Methods

1. Page 6: What is known about the makeup of the 5% of young people not captured in the Scottish annual pupil census? could it be that people with intellectual disability and complex needs, who may be at higher risk of death, and who have different causes of death, are more likely to be in the 5% that is not surveyed?

- *Response - We acknowledge that there are children in the Scottish population who will not be listed in our study. The pupil census does include additional grant-funded placements in independent schools (where some children with intellectual disabilities and complex needs are placed), and children in state care, and 'looked after children'. However it does not cover home-educated or privately educated children in Scotland. There is limited information available on the number of children educated in independent schools. According to the Scottish Council of Independent Schools which covers 74/102 of independent schools in Scotland, member schools include between 29-30,000 pupils in Scotland, approx. 4.1% of the entire school-age population. We do not know if children with intellectual disabilities are more likely to be privately educated (i.e. family pay the fee), but have no reason to believe that they are.
- While it is not currently possible to estimate the number of people with intellectual disabilities who are home-educated, a 2018 BBC survey found an estimated that 0.1% of all Scottish children were home-educated[1]. We do not know if children with intellectual disabilities are more likely to be home-educated, but have no reason to believe that they are, however, parents of children with exceptional healthcare needs or children with life-limiting conditions receiving palliative care may be missing from our study.
- We have added the following to the text:

Discussion, page 19: "Additionally, while we believe this population to be highly representative of children with intellectual disabilities across Scotland, we acknowledge that we were unable to access data on children not in school; there may be some under-ascertainment of children with intellectual disabilities with exceptional and complex health needs unable to attend school."

2. Page 7: Why were non-singleton births excluded? this seems unnecessary and is unexplained.

- *Response - The record linkage was carried out using probabilistic record matching using date of birth, sex and postcode. The highest linkage score was used for each record. Study participants that were not successfully linked were not included in the study. Due to the lack of full names used, it was not possible to distinguish between duplicate census records for same sex siblings residing at the same postcode, so non-singleton births were identified using maternity records and excluded.
- We have updated the methods to provide more detail on the record linkage:

Methods, page 5 "The record linkage methodology required date of birth, sex and postcode, however since names were not used to link pupil records to the health data, we excluded non-singleton births (available for Scottish-born pupils only, identified from linkage to maternity records). Unlikely matches were excluded and the most likely match was selected as the correctly linked pupil record. We also excluded any records with duplicate pupil records or where the linkage was tied with another patient."

- Furthermore, after receiving feedback from peer reviewers, we have identified a small number of pupil records where linkage to healthcare database was tied to several patients. We have now excluded these records, and removed the patients from our analysis. This resulted in exclusion of n=317 people from our intellectual disability group (or 1.7%) and n=3950 (<1%) from our control group, who had tied linkage to another patient's healthcare records. These have subsequently been removed, and as such our analysis has been updated.

3. The database would appear to be able to distinguish between support related to an autism specific issue or intellectual disability. it is therefore unclear why people with autism without intellectual disability were not included in the control sample.

- *Response - The focus of this study was mortality of children and young people with intellectual disabilities, not any of the other types of additional support needs.

- Additionally, we have changed the methods section, page 5 -“Pupils with solely autism were also excluded from controls, to eliminate potential mislabelling of support need for either autism or learning disability in the absence of clinical diagnoses.”

4. The concept of immortal time bias may be unfamiliar to some readers, as it was a new term to me.

- *Response - We have made the text clearer on immortal time bias in the study. The text now reads: Methods, page 6: “Since only those pupils who attended school in at least two years over our observed study period were eligible, the period between the first and second record introduced an immortal time bias, where no deaths could have occurred, and therefore the entry to the study was defined as the date of their second pupil census record.”

5. I was interested in why some of the data is presented as childhood (5-14) and young people and other data is presented as 5-year age bands. it might be useful to be consistent across the analyses.

- *Response - For both the childhood SMR and young people SMR, the mortality rates were age-standardised indirectly using the expected rates from each one-year band. However due to low sample size for the cause-specific rates, observed deaths within each one-year age-band were too low to use, and included zero deaths in some one-year age bands and so we further collapsed ages into 5 year age bands. Stratification of results by children and young people [into aged 5-14, and 15+] was also no longer possible. We therefore presented ratios of all the deaths across the entire study group, and used 5-year age-bands instead of one-year age-bands, to reflect the low numbers of deaths.

- We have amended the methods section, page 6 :“The mortality rates were indirectly standardised for both males and females using the expected age-specific mortality rates per one-year age-group, using STATA’s “strate” command, to calculate age- and sex-standardised mortality ratios (SMRs) for pupils with versus without intellectual disabilities. The expected rates were derived from the comparison group age and sex-specific rates. The SMRs were subsequently calculated stratified by age, into childhood (aged 5-14 years) and young people (aged ≥15 years), and by sex [...] For cause-specific SMRs, indirect age-standardisation was also performed, but using expected rates per 5-year age-bands to age-standardise rates.”

Results

6. Page 11: the statement regarding the minimum number of deaths in particular categories that were reported seems to vary from 10 to 20. Later in the Results section of the manuscript, there is also reference to a minimum cell size of 5. Clearer disambiguation may be helpful for the reader as I was uncertain about these distinctions.

- *Response - We agree that there is ambiguity in how low samples are classified as unreliable.

Throughout the study, n=10 has been used as the minimum required number of deaths needed to calculate rates and rate ratios. We have now labelled all mortality rates based on between 10 and 20 deaths as unreliable due to low numbers of deaths.

- In the methods section, page 7, the text has been amended to: “For categories which had fewer than ten deaths, no calculation was attempted due to lack of reliability in the small number of deaths. Furthermore, in keeping with the ONS mortality methodology[22], all mortality rates based on between ten and twenty deaths were labelled as unreliable.”.

- Figure 2 has been also been modified to include the label “U” for unreliable rates. The footnote has been amended: “SMRs which were calculated using low numbers (between 10 and 20 deaths) are labelled “U” as unreliable”

7. Page 14: there may be value in documenting the change in SMR for people without intellectual disability once external causes of death were excluded? A comment on the relative magnitude of the change in males and females with intellectual disability once external causes of death were excluded might also be warranted. Though arguably, this could be left to the discussion.

- *Response - We have added the increase in magnitude for the different SMRs by gender to emphasise the change upon exclusion of deaths due to external causes. The text now reads:

Results, page 10: “Exclusion of external causes of death resulted in a considerable increase in the all-cause SMR for both females and males with intellectual disabilities; overall SMR was 21.6 (17.8, 26.3), female SMR 25.6 (18.8, 34.9) versus male SMR 19.6 (15.3, 25.2). This produced a relative increase of 10 more deaths overall for pupils with versus without intellectual disabilities, which was similar in females (+9.0 increase), and males (+9.8 increase).”

8. Page 15: the last part of the sentence at line 19 “there were also differences in the most common or contributing causes of death- those with any mention on the death certificate” is not necessary as this has been clearly stated in the methods section.

- *Response - this sentence has been shortened: “There were also differences in the most common all-contributing causes of death”.

9. Rather than by chapter number, Table 3 may be better organised according to most frequent causes of death in one or other of the cohorts.

- *Response - Table 2 (previously Table 3) has now been updated in order frequency of all-contributing causes of death in pupils with intellectual disabilities.

Conclusion/future direction

10. : Future studies could consider looking at predictors of death in children and young people to inform translation of findings into clinical benefit for people with ID.

- *Response - We agree with this important point and have added this sentence to our conclusion, page 20.

Ethical approval

11. It is usual to quote ethics committee approval numbers in my jurisdiction. Should this be included?

- *Response - We have added our Public Benefit and Privacy Panel approval number for the study to the manuscript, page 21.

Reviewer 2 Comments

1. page 7, line 35 :except where there were (for) fewer than...

- * Response - This typo has been amended.

Reviewer 3 Comments

Introduction

1. The introduction provides good background and justification for the study. As Table 1 presents more detailed information from previous studies only, it may be more appropriate to include this in online appendix rather than the main body of the paper.

- *Response - Table 1 has now been added as supplementary data (Table S1).

Methods:

1. The decision to include non-singleton births could be more clearly justified.

- *Response - The record linkage was carried out using probabilistic record matching using date of birth, sex and postcode. The highest linkage score was used for each record. Study participants that were not successfully linked were not included in the study. Due to the lack of full names used, it was not possible to distinguish between duplicate census records for same sex siblings residing at the same postcode, so non-singleton births were identified using maternity records and excluded.

- We have updated the methods to provide more detail on the reason why non-singleton births were excluded:

Methods, page 5: “The record linkage methodology required date of birth, sex and postcode, however since names were not used to link pupil records to the health data, we excluded non-singleton births (available for Scottish-born pupils only, identified from linkage to maternity records).”

2. Although the authors state when follow-up started for each child, they do not state when it ended. As the authors were interested in deaths in children/ young people and they criticise previous work for inclusion of adult deaths, did they censor patients / end follow-up at age 19?

• *Response - We agree and have now added:

Methods, page 5: "Pupils were also censored upon reaching aged 25 if they reached this age during the observation period, so that the maximum follow-up age was 24 years old."

• Additionally we have added the age-group breakdown of the study population to Table 1.

3. The data linkage process is not entirely clear based on the information provided. Although the linkage was done by a third party by National Services Scotland the authors should still be able to report if linkage was based on exact or probabilistic matching, the variables on which linkage was carried out and how inexact matches were handled.

• *Response - We have further updated the methods section about the record linkage as follows:

Methods, page 5: "Unlikely matches were excluded and the most likely match was selected as the correctly linked pupil record. We also excluded any records with duplicate pupil records or where the linkage was tied with another patient."

• Furthermore, after receiving feedback from peer reviewers, we have identified a small number of pupil records where linkage to healthcare database was tied to several patients. We have now excluded these records, and removed the patients from our analysis. This resulted in exclusion of n=317 people from our intellectual disability group (or 1.7%) and n=3950 (<1%) from our control group, who had tied linkage to another patient's healthcare records. These have subsequently been removed, and as such our analysis has been updated.

4. Chapters for ICD10 codes R00-R94 and R95-R99 are mentioned in the methods but then subsequently referred to as Chapters 18 and 19 in the results section - to aid the reader it would be preferable if the description was consistent.

• *Response - We have changed the methods section, page 7, to "All deaths where the underlying cause was ill-defined; defined by ICD 10 WHO guidelines[20] as codes in Chapter 18 excluding R95, were also re-classified as "unknown".

Results

1. There is some repetition of results given in both the text and tables. The authors may wish to consider condensing some of this material and/or moving some results to an online appendix.

• *Response - We have re-examined our results section to check for repetition throughout our tables, and altered the text to reduce repeated information.

• On page 10, [for all-contributing causes ICD chapter 19]... "injury, poisoning and other consequences of external causes in 219 of their 461 deaths (compared with 10 of the 106 deaths in the pupils with intellectual disabilities)." has now been removed.

• On page 16, [For underlying causes, this was only possible for the two largest categories (by ICD 10 chapters);]... "SMR 101.4 (67.4, 152.5) for congenital malformations, deformations and chromosomal abnormalities, and SMR 89.7, (64.4, 125.0) for diseases of the nervous system." has been changed to "congenital abnormalities, and diseases of the nervous system."

• Tables 2 and 3 have been condensed into one single table, now Table 2.

Ethical approval

1. The authors should provide the review number/ approval number for the ethical approval received.

• *Response - We have added our Public Benefit and Privacy Panel approval number for the study to the manuscript, page 21.

Reviewer 4 comments

Methods

1. While I totally understand the exclusion of non-singleton births, I wonder why only pupils with intellectual disabilities recorded in at least two different school years were included in the intellectual

disabilities group. Does this mean that those that had only recorded intellectual disability in one school year were excluded from the whole study?

I also understand that those that had only one-year record of no intellectual disability were also excluded. Was then an inclusion criterion to have at least two records over time to be part of the sample? If this is the case? What do we know about those that have only one record?

I can understand this a strategy to ascertain that they are correctly identified as individuals of the intellectual disabilities group, but why not just run sensitivity analyses comparing those with 1 school year and those that were at least in two years? And same with comparison group? Can the authors please provide some numbers about how many were excluded for this reason and how might they differ from the sample that was considered in the study. This is particularly relevant to understand to what extent these results are generalizable.

- *Response – The reviewer is correct that we excluded from the study all pupils with only one record of intellectual disabilities support as well as everyone who only appeared once in the census.
- Overall there were 7,135, pupils with a single year of support due to intellectual disabilities who were excluded from the study, who had ≥ 2 census records – and a further 1,727 pupils with intellectual disabilities support who appeared only once in the census (8,862 pupils in total excluded). There were a total of 131,776 control pupils who were also excluded because they appeared in only a single census year.
- We have now reported this information to in the results:
- Results, page 8 – “there were 27,140 pupils who had ever registered as having an Additional Support Needs due to intellectual disabilities, and of these, 18,278 (1.9% of pupils) met the criteria of having in at least two records of support. The remaining 8,862 pupils with a single support record were excluded.
- Results, page 8: “Of these, 131,776 were excluded due to appearing in only one year of the census.”
- We have also now conducted and included a sensitivity analysis as part of this study – and reported mortality results for pupils with at least one record of intellectual disability compared to controls with at least one census record. Pupils who were not included in the main analysis had significantly fewer disabilities adaptations, and higher numbers of pupils in this group also had years without intellectual disabilities support. The mortality ratio dropped from 11.6 to 9.5. The text has been modified as follows:
Methods, Page 7: “A sensitivity analysis was carried out using wider inclusion criteria from the education data for both groups; the intellectual disability group included all pupils with at least one record of support at school due to intellectual disabilities. The control group included all pupils with at least one census record, and without support records for intellectual disabilities or autism. There were no other methodological changes made to age standardising process or censor dates, but entry date was changed to the date of the first record of support need for pupils with intellectual disabilities or the first census date for pupils without intellectual disabilities.”
- Results, Page 16:
• Of the 27,140 pupils with at least one record of support due to intellectual disabilities, 65% were male, and compared to the main analysis group, there were significant reductions in frequency of school adaptations (physical disability reduced from 11% vs 9%, ($p < 0.001$), curriculum adaptations from 36% to 31% ($p < 0.001$), and communication adaptations from 19% to 16% ($p < 0.001$). There were higher numbers of pupils in this group with years without intellectual disabilities support. There were 156 deaths in the intellectual disabilities group (134 per 100,000 person-years [114.2, 156.3]) compared to 684 deaths (13.8 per 100,000 ([12.8, 14.8]) amongst the control group. The SMR for this sensitivity analysis was 9.5 (95% CI 8.1, 11.1), a change of minus 2 excessive deaths compared to the main analysis SMR. Mean age of death was similar in the sensitivity group, being 14.4 years (13.7, 15.1) in the intellectual disabilities group, and 16.2 (15.9, 16.5) in the control group. The ratio of deaths by sex were also very similar, with no difference for the intellectual disabilities group; 61% deaths were in males, similar to the proportion of males in the group ($p = 0.306$), and an increase in

male deaths amongst controls; 63% of deaths were in males, whereas only 50% of the control group were male ($p < 0.001$)."

2. Were there any individuals that had two years or more with intellectual disabilities and two years or more without? If yes, how were these handled?

- *Response - We found that there were pupils who met the inclusion criteria who still appeared in other census years with no support and we have included a breakdown of this in the study under the heading "missing years of support" for the main analysis:
- Results, page 8: "There were 11,329 pupils (62%) of the intellectual disability group who appeared in certain census years without having a record of support. The majority, 70%, ($n=7,970$) were before the accrual of the first record; these pupils had a median 2 pupil census records prior to receiving their support (interquartile range (1,3). There were also 3,359 pupils or 18% of the entire study group who went on to have census records without support records, after receiving intellectual disability support provision. These pupils had a median 1 subsequent year (IQR 1,2) of no support, out of a median 4 remaining years (IQR 3,6) in the census."
- As described in the text, the main reason for census years without support for the intellectual disability group was a delay to "diagnosis" - for 70% of pupils who had missing support records. There were approximately 18% of the study group who upon "diagnosis" went on to have years with no support. This rose to 26% for the sensitivity analysis group with intellectual disabilities.

3. About the follow up and the modelling approach: Education data was extracted between 2008 and 2013 (that is, over 5 years) and we understand that deaths were from 2008 to February 2015 in Scotland. Is this correct? Then there might be a lot of variability in length of follow up across individuals? If yes, how did the authors deal with this?

- *Response - The first pupil census available was September 2008, and due to requirement of at least two years in the census, the earliest entry into the follow up period was from 2009. Death records were followed up to February 2015. We have added more detail to the methods regarding definition of study entry and censor date:

Methods, page 6: "the period between the first and second record introduced an immortal time bias, where no deaths could have occurred, and therefore the entry to the study was defined as the date of their second pupil census record."

Methods, page 6: "Crude mortality rates were calculated using the censor date, 13 February 2015 or date of death."

- There was variability between the two groups but the process of age-standardisation produces a weighted rate based on person-time at risk per one year age-band, and adjusts for variability in age and differences in the length of follow up between the two groups being compared. We have added more information to the methods section, page 6: "the mortality rates were indirectly standardised for both males and females using the expected age-specific mortality rates per one-year age-group, using STATA's "strate" command, to calculate age- and sex-standardised mortality ratios (SMRs)."

4. I am missing more details on the statistical approach followed. More information on the modelling approach, how the models were built, whether robust approaches for smaller samples (in the case of mortality cause) were considered? Where the assumptions checked? Some survival graphs will also contribute to a better understanding of the results. All this information would be also useful to explore other limitations that might have not been mentioned.

- We have used indirect standardisation and assumed observed deaths assumed to vary according to the Poisson distribution. The mean and variance were checked and found to be almost identical. We have included further statistical information including assumptions to the methods:
- Methods, page 6: "For indirect standardisation, observed deaths were assumed to be independent and vary with the Poisson distribution. The mortality rates were indirectly standardised for both males and females using the expected age-specific mortality rates per one-year age-group, using STATA's "strate" command, to calculate age- and sex-standardised mortality ratios (SMRs) for pupils with versus without intellectual disabilities. 95% confidence intervals were calculated based on the

quadratic approximation of the log likelihood. Expected rates were calculated using fixed age and sex-specific rates, from the large control population.”

- We have now added Kaplan-Meier survival curves and for the main analysis also reported the Cox-proportional hazards ratio (the KM plots are reported in the supplementary information). For cause-specific mortality we have re-calculated SMRs employing robust estimators. We have added the following text to the methods and results sections:

Methods page 6: “For all-cause mortality, Kaplan-Meier survival curves were plotted for the overall time period for both groups. Cox-proportional hazards models are also presented, adjusted for age and sex.”

Results page 10 “The Cox-proportional hazards ratio for all-cause mortality, adjusted for age and sex, was found to be very similar; HR : 11.97 (9.64, 14.86). Proportional hazards assumption was met (p=0.4217). Kaplan-Meier survival curves for the overall time period are found in online supplementary data (Appendix S2).”

Methods page 7: “For cause-specific SMRs [...] robust standard errors were used.”

- Furthermore, we have also now labelled mortality rates based on between ten and twenty deaths as unreliable.

Methods, page 7 : “For categories which had fewer than ten deaths, no calculation was attempted due to lack of reliability in the small number of deaths [...] in keeping with the ONS mortality methodology[22], all mortality rates based on between ten and twenty deaths were labelled as unreliable.”

- In the overall limitations (page 2), we have mentioned how low numbers of cause-specific mortality rates limited further analysis.

Minor

1. Table 2 refers to baseline data which we assume is 2008. It would help that is clear in the title of the table. In the abstract there is no sentence summarizing the statistical approach.

- *Response: We agree and have made changes to the text on page 6 (methods)– “Non-modifiable descriptive data on sex, ethnicity and SIMD, were taken from each pupils’ first year in the census. For disability requirements, all records across multiple pupil census years were used to define whether having ever received adaptation requirements”.

- We have also made changes to the text on page 7 (results)-“Using data from the pupils’ first year in the Census”

- We have added a footnote to Table 1 -“Data taken from first census record, except for disability adaptation any record across census years”

References

1. BBC survey :BBC News 26 April 2018. <http://www.bbc.co.uk/news/uk-england-42624220>

VERSION 2 – REVIEW

REVIEWER	Julian Trollor University of New South Wales, Department of Developmental Disability Neuropsychiatry
REVIEW RETURNED	26-Apr-2020

GENERAL COMMENTS	Thank you. The authors have addressed all of the issues raised. I have no further comments.
---

REVIEWER	Jenny Bourke Telethon Kids Institute, Australia
REVIEW RETURNED	16-Apr-2020

GENERAL COMMENTS	As noted in my original review this study broadens the current literature in the area of mortality in children and young people with ID and importantly identifies the inequalities that exist. Whilst I found little correction needed in the original manuscript I feel the comments from other reviewers have helped to improve the paper and feel it is a very worthwhile publication.
--

REVIEWER	Sarah Lay-Flurrie University of Oxford, UK
REVIEW RETURNED	27-Apr-2020

GENERAL COMMENTS	All my previous comments have been addressed.
---